rsob.royalsocietypublishing.org

**Subject Area:**
neuroscience

neuropathic pain, spinal cord, microglia, purinergic signalling, P2X4 receptor

**Author for correspondence:**
Kazuhide Inoue
e-mail: inoue@phar.kyushu-u.ac.jp

# A state-of-the-art perspective on microgliopathic pain

Kazuhide Inoue

Department of Molecular and System Pharmacology, Graduate School of Pharmaceutical Sciences, Kyushu University, Fukuoka 812-8582, Japan

(iD) KI, 0000-0003-0385-9577

Acute nociceptive pain is an undesirable feeling but has a physiological significance as a warning system for living organisms. Conversely, chronic pain is lacking physiological significance, but rather represents a confusion of nerve functions. The neuropathic pain that occurs after peripheral nerve injury (PNI) is perhaps the most important type of chronic pain because it is refractory to available medications and thus remains a heavy clinical burden. In recent decades, studies have shown that spinal microglia play a principal role in the alterations in synaptic functions evoking this pain. It is also clear that the P2X4 receptor (P2X4R), a subtype of ionotropic ATP receptors, is upregulated exclusively in spinal microglia after PNI and plays a key role in evoking neuropathic pain. Neuropathic pain is caused by several conditions associated with activated microglia without nerve damage. 'Microgliopathic pain' is a new concept indicating such abnormal pain related to activated microglia.

## 1. Introduction

Neuropathic pain is evoked by the damage of pain-related neurons in traumatic injury, diabetes mellitus, autoimmune diseases, infection or bone compression in cancer. Neuropathic pain seems one of the most debilitating forms of chronic pain, and is a significant clinical problem because it is refractory to medications such as opioids and nonsteroidal anti-inflammatory drugs [1]. More than 20 million patients worldwide suffer from neuropathic pain. Previously, nearly all researchers believed that neuropathic pain was the direct result of altered functions in the peripheral and central nervous systems after peripheral nerve injury (PNI). As many scientists reported studies focused on neurons to understand the mechanism of neuropathic pain, they suggested that neuropathic pain is a result of abnormal excitability of the secondary sensory neurons in the dorsal horn, and that histological reorganization of pain signalling in the peripheral and central nervous systems causes neuropathic pain [1,2]. They also proposed many pharmacological tools against molecular targets in neurons to treat this pain. However, these tools, including candidate compounds, did not produce significant therapeutic effects in patients [3]. The failure of compounds targeting neurons suggested that non-neuronal mechanisms may be involved in neuropathic pain. Recently, a growing body of evidence shows that spinal microglia activated in response to PNI have important pathophysiological roles in the modification of synaptic transmission of pain [4–8]. This review describes recent advances in understanding the mechanism of evoking neuropathic pain through the functions of P2X4Rs in spinal microglia after PNI.

## 2. Normal acute pain

Under normal conditions, acute nociceptive pain has a physiological significance as a warning system that enables the detection of danger signals that threaten the homeostasis of living things. Painful stimuli evoke action potentials in the distal ends of C-fibres or Aδ-fibres belonging to dorsal root ganglion

rsob.royalsocietypublishing.org Open Biol. **8**: 180154

(DRG) neurons. These spikes conduct to the central ends of these DRG neurons and transmit to secondary sensory neurons in the dorsal horn superficial layer through mainly glutamatergic synaptic transmission, and finally to the sensory cortex, evoking a pain sensation. Conversely, touch stimuli evoke action potentials in Aβ-fibres belonging to DRG neurons, and these spikes are transmitted to the sensory cortex resulting in touch sensation. There are no reports indicating overlap of these sensory inputs under normal conditions. In other words, light touch stimuli do not cause pain sensation but inhibit pain signalling under normal conditions, as described below. Action potentials evoked by touch stimuli in Aβ-fibres partly transmit to inhibitory interneurons, resulting in the release of the inhibitory neurotransmitters, γ-aminobutyric acid (GABA) or glycine. GABA acts at secondary neurons to evoke hyperpolarization and the inhibition of pain signalling (figure 1). However, under pathophysiological conditions, there is ample evidence showing that touch stimuli evoke strong pain sensations, as described in the next section.

## 3. Neuropathic pain and microglial activation after PNI

Chronic pain lacks physiological significance as a warning system. Neuropathic pain that occurs after PNI is the most important type of chronic pain because effective medications are lacking, and thus it remains a significant clinical burden. The symptoms of neuropathic pain include spontaneous pain, hyperalgesia and tactile allodynia. Tactile allodynia is painful hypersensitivity to normally innocuous touch stimuli and is an interesting shift in pain sensation elicited by touch stimuli that are not seen in normal physiological conditions.

In 2003, a new concept of evoking neuropathic pain was proposed where spinal microglia are activated after PNI, and P2X4Rs on these activated microglia have an important role for eliciting neuropathic pain [9]. An initial observation that tactile allodynia after PNI is reversed by an inhibitor of P2X4Rs in the spinal cord led to this concept [9]. It was also shown that the expression of P2X4Rs in the spinal cord is upregulated exclusively in microglia after PNI and that animals with P2X4R knock-down or knock-out in the spinal cord are resistant to tactile allodynia [9–11]. These results indicate that PNI-induced allodynia depends on microglial P2X4R signalling [9]. This report brought glial pain mechanisms to the forefront of research. Much research was subsequently performed, and the following findings became clear. First, various lines of evidence indicated that injured neurons might induce the activation of microglia in the dorsal horn [12,13]. Although it is currently unclear which factors are essential in the activation of microglia, cytokine interferon (IFN)-γ and platelet-derived growth factor (PDGF) may be candidates for activation factors evoking microglial activation. Another study showed that IFN-γ levels are increased in the spinal cord after PNI [14] and that spinal microglia exclusively express a receptor for IFN-γ (IFN-γR) in naive animals, and this receptor stimulation changes microglia into an activated form. Injection of IFN-γ produces long-lasting tactile allodynia, and microglial activation and tactile allodynia after PNI are severely impaired by ablating IFN-γR [15]. Moreover, it was found that spinal microglia show the upregulation of Lyn tyrosine kinase and

P2X4R by IFN-γ-stimulation [15]. These results suggest that IFN-γR is an important molecule in the activation of microglia and neuropathic pain.

PDGF expressed in dorsal horn neurons plays a role in evoking neuropathic pain after PNI [16]. It was also reported that spinal microglia are involved in PDGF-evoked tactile allodynia [17]. Specifically, intrathecal injection of the PDGF B-chain homodimer (PDGF-BB) in naive rats evoked long-lasting tactile allodynia in a dose-dependent manner. The immunofluorescence for the phosphorylated PDGF β-receptor (p-PDGFRβ, an activated form of this receptor) was markedly increased by PDGF injection into dorsal horn microglia. After treatment with PDGF-BB, microglial cell numbers and morphology indicated that microglia are modestly activated. In addition, intrathecal administration of minocycline (an inhibitor of microglial activation) inhibited PDGF-BB-induced tactile allodynia [17]. These results suggest that PDGF evokes the activation of spinal microglia and tactile allodynia.

## 4. Mechanism of evoking mechanical allodynia

In 2005, it was reported that stimulation of microglial P2X4Rs evokes the synthesis and release of brain-derived neurotrophic factor (BDNF) [10,18]. It was also shown that BDNF might downregulate the function of the KCC2 chloride transporter through activation of transmembrane tyrosine kinase B (TrkB) in lamina I secondary neurons, resulting in a depolarizing shift of the anion reversal potential ($E_{anion}$) [19]. This shift inverts the polarity of currents activated by γ-amino butyric acid (GABA) and glycine, such that GABA and glycine cause depolarization, rather than hyperpolarization, in these secondary sensory neurons [19]. Moreover, it was reported that ATP stimulation evokes the release of BDNF from microglia *in vitro*, and BDNF injection mimicked the alteration in $E_{anion}$ in lamina I secondary neurons in an *in vivo* study. Inhibition of the interaction between BDNF and the TrkB receptor reverses allodynia, and both nerve injury and the injection of ATP-stimulated microglia into the dorsal horn cause a similar shift of the $E_{anion}$ [19]. These studies suggest that microglial P2X4Rs are central players in the pathogenesis of the allodynia associated with neuropathic pain. However, there are no data explaining where the ATP that stimulates P2X4Rs originates.

Though extracellular ATP in the spinal dorsal horn stimulates microglial P2X4Rs, resulting in neuropathic pain after PNI, the type of cells and the mechanism for ATP release within the spinal cord has remained a mystery. Recently, it was reported that the vesicular nucleotide transporter (VNUT) in dorsal horn neurons is a key molecule for ATP release and neuropathic pain [20]. In that report, VNUT expression and extracellular ATP content ([ATP]e) in the dorsal horn increased in proportion to pain hypersensitivity in wild-type mice after PNI [20]. Furthermore, the increase of [ATP]e and the tactile allodynia of neuropathic pain were prevented in VNUT-deficient mice, or following treatment with exocytosis inhibitors. Similar suppression of tactile allodynia and spinal [ATP]e was found in mice bearing a specific deletion of VNUT exclusively in dorsal horn neurons. Notably, any suppressions of tactile allodynia and spinal [ATP]e were not seen in a specific deletion of VNUT exclusively in primary sensory neurons, microglia or astrocytes

rsob.royalsocietypublishing.org  *Open Biol.* **8**: 180154

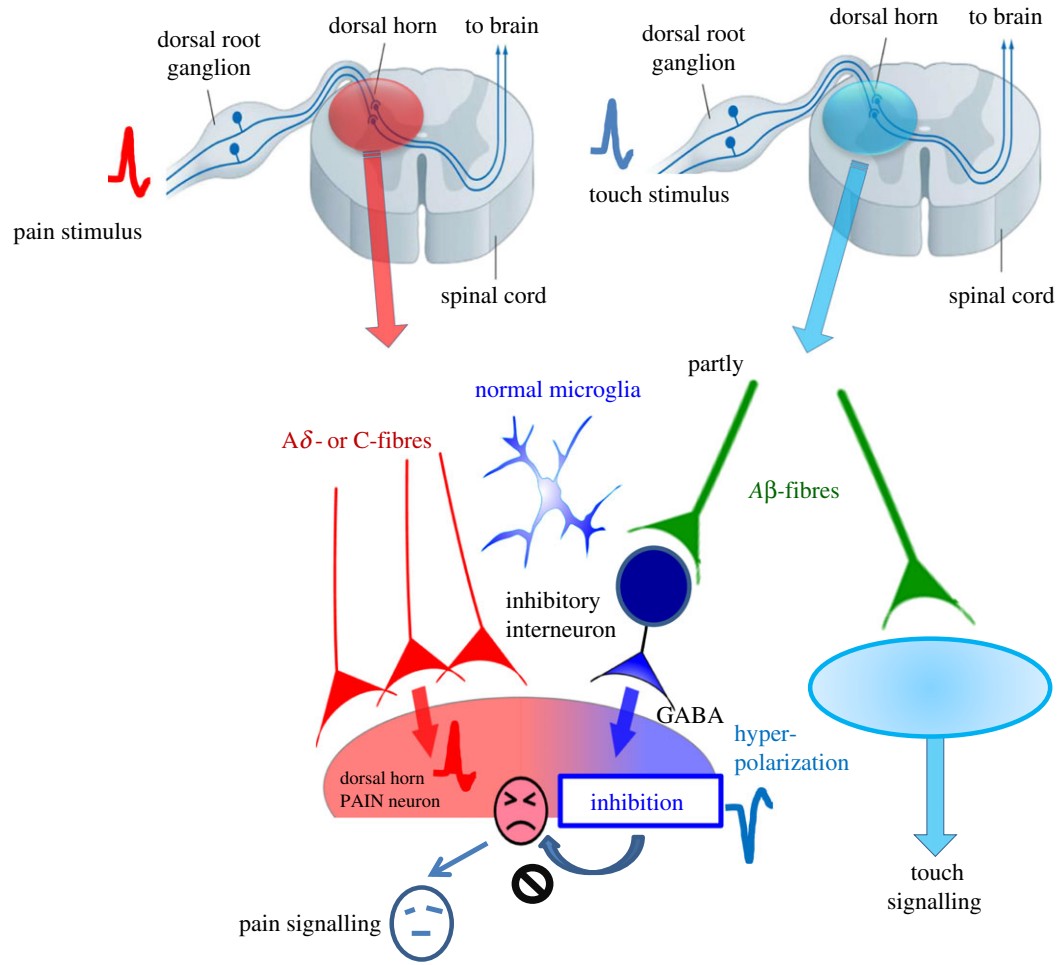

**Figure 1.** Pain signalling under normal conditions. Under normal conditions, painful stimuli evoke action potentials in dorsal root ganglion neurons (C-fibres or Aδ-fibres). These spikes transmit to secondary sensory neurons (pain neurons) in the dorsal horn and finally to the cortex. Touch stimuli evoke action potentials in $A\beta$ fibres that partially transmit to inhibitory interneurons, resulting in the release of γ-aminobutyric acid (GABA). GABA acts as an inhibitory transmitter to secondary neurons, resulting in hyperpolarization and the inhibition of pain signalling.

[20]. Of the various types of dorsal horn neurons, inhibitory interneurons seem to play a crucial role in the release of ATP. Such release results in PNI-induced allodynia because ATP release from spinal cord slices was suppressed in mice selectively lacking VNUT in vesicular GABA transporter (VGAT)-expressing inhibitory neurons generated by crossing VNUT-floxed ($Slc17a9^{fl/fl}$) mice with *Vgat-Cre* mice (*Vgat-Cre;Slc17a9*$^{fl/fl}$) [21]. These data indicate that the VNUT-dependent release of ATP from GABAnergic inhibitory neurons of the spinal dorsal horn is important for the production of tactile allodynia in neuropathic pain. Thus, it is speculated that touch stimuli evoke action potentials in Aβ-fibres of DRG neurons and these spikes transmit to inhibitory interneurons resulting in the release of ATP as well as GABA. In succession, ATP stimulates the microglial release of BDNF which affects secondary sensory neurons, resulting in a depolarizing shift of $E_{anion}$. GABA released from inhibitory interneurons stimulates secondary neurons to evoke depolarization which transmits to the sensory cortex, resulting in pain sensation (figure 2).

There is one technical problem in this hypothesis. To assess mechanical allodynia, calibrated von Frey filaments (0.4–15 g, Linton Instrumentation, Diss, Norfolk, UK) are applied to the plantar surface of the hind paws of animals. The 50% paw-withdrawal threshold (PWT) is determined by the up–down method. Scientists have used this method for pain studies for more than 100 years, and they speculate

that very light filaments cause only touch sensation through Aβ-fibres. However, there is no evidence for this speculation. Specific tools to manipulate Aβ-fibre function in awake, freely moving animals are required for clarifying this issue. Recently, it was reported that illuminating the plantar skin of transgenic rats in which light-activated channels (channelrhodopsin-2; ChR2) are expressed only in Aβ-fibres elicits pain behaviours after PNI that are very similar to pain behaviours confirmed by von Frey testing with similar time-courses after PNI [22]. In these rats, illumination of the skin after PNI increased the number of the activity markers c-Fos and phosphorylated extracellular signal-regulated protein kinase (pERK) in spinal dorsal horn Lamina I neurons [22]. In addition, optogenetic Aβ-fibre stimulation after PNI caused excitation of Lamina I neurons in whole-cell recording, which were normally silenced by this stimulation without PNI [22]. These data indicate that optogenetic activation of primary afferent Aβ-fibres in PNI rats produces excitation of Lamina I neurons and neuropathic pain.

## 5. Microgliopathic pain

Neuropathic pain is caused by several conditions after PNI, as mentioned above. Recently, several reports indicated that abnormal activated microglia are able to cause neuropathic pain even in in naive animals [15,17,19,23–25]. From these

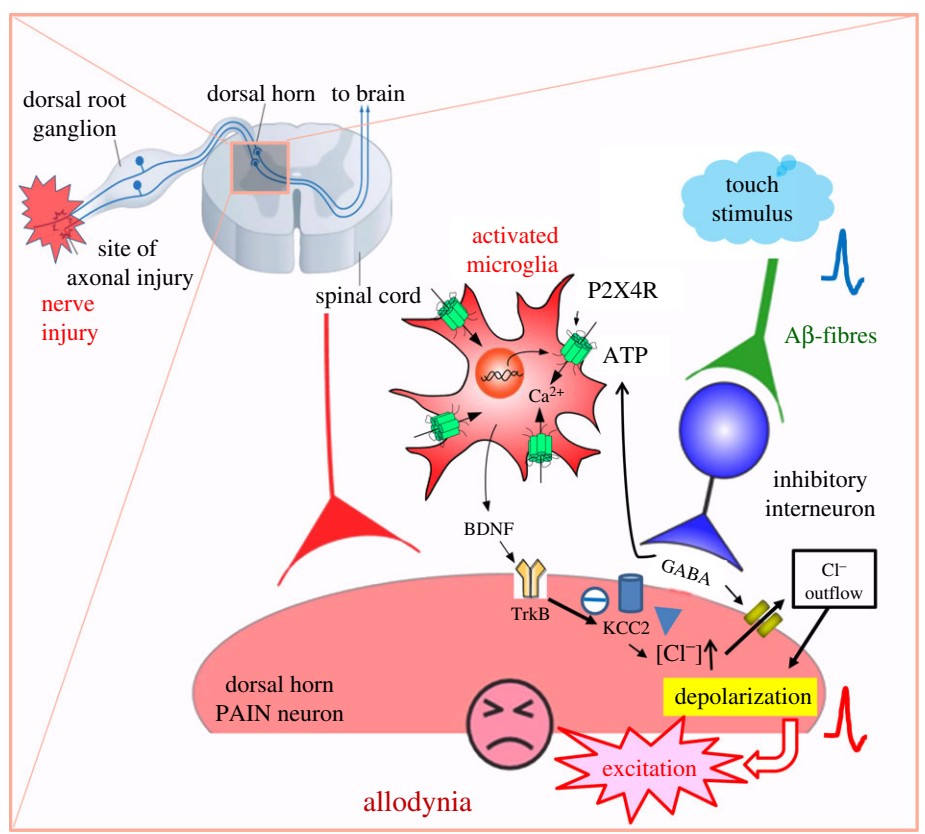

**Figure 2.** A hypothesis for the elicitation of allodynia through P2X4 receptor stimulation in activated microglia following a peripheral nerve injury (PNI). Nerve injury activates dorsal horn microglia to overexpress P2X4Rs. ATP from dorsal horn neurons stimulates microglial P2X4Rs to cause the release of brain-derived neurotrophic factor (BDNF). BDNF stimulates TrkB receptors located on lamina I secondary sensory neurons in the dorsal horn to cause downregulation of the KCC2 potassium-chloride transporter, resulting in increased intracellular $[Cl^-]$ and a depolarizing shift in the anion reversal potential ($E_{\text{anion}}$). In this pathological condition, touch stimulation causes GABA release from inhibitory interneurons. GABA opens $Cl^-$ channels of lamina I secondary neurons and leads to $Cl^-$ outflow, resulting in depolarization, rather than hyperpolarization of these neurons to evoke action potentials that reach the cortex. In this way, innocuous touch stimulations are converted to pain sensation as allodynia.

findings, we have created the new concept named 'microgliopathic pain' that indicates abnormal pain derived by activated microglia without accompanying nerve damage.

Even in normal states, microglia are quite active in communicating with other cells and receive signals from the outside environment. This observation led to the hypothesis that the surveillance mode of microglia may be shifted to a reactive state by excess stimulation by signals from the outside environment. This shift will depend on the strength of the signals and eventually will return to a normal state. After PNI, the microglia shift completely towards a reactive state, and various abnormal phenotypes are expressed, resulting in a prolonged state of pain hypersensitivity. Such a phenotype would include high levels of P2X4R expression, which activates the BDNF–TrkB–KCC2 pathway mentioned above, the Cathepsin S inducing nucleotide-binding oligomerization domain (NOD)-, the C-terminal leucine-rich repeat (LRR)- and pyrin domain-containing protein 3 (NLRP3) inflammasomes, or the fractalkine–CX3CR1–p38MAPK–IL-1$\beta$ or TNF$\alpha$ signalling pathways. An important question is what types of core molecules control the expression of such abnormal phenotypes. It has been reported that some transcription factors (IRF8 and IRF5) may be involved in this regulation [26,27].

IRF8 is a member of the IRF family (IRF1–9) expressed in immune cells such as lymphocytes and dendritic cells [28].

Several reports indicate that IRF8 acts as a transcription factor in microglia [26,29–31] and is critical for microglial activation and neuropathic pain [26]. Furthermore, the IRF8 expression is markedly enhanced exclusively in microglia in the spinal cord after PNI [26]. IRF8 expression is upregulated as early as day 1, peaks on day 3 and the upregulation persists for at least several weeks after PNI. PNI-induced tactile allodynia was not detected in IRF8-deficient mice. Intrathecal injection of a small interfering RNA (siRNA) targeting IRF8 inhibited the upregulation of spinal IRF8 and allodynia after PNI in normal mice. These findings indicate that activation of IRF8 is a sustained event after PNI in spinal microglia. In both *in vitro* and *in vivo* studies, IRF8 promoted the transcription of P2X4R and the innate immune response toll-like receptor 2 (TLR2), the CX3CR1 chemokine receptor, interleukin-1$\beta$, cathepsin S, P2Y12R and BDNF, which are important factors involved in neuropathic pain [26].

IRF5 directly controls the transcription of P2X4R in microglia after PNI [27]. Importantly, IRF5 is an IRF8-regulated gene and increases in spinal microglia in an IRF8-dependent manner after PNI. Furthermore, fibronectin stimulates the translocation of IRF5 from the cytoplasm into the nucleus, resulting in IRF5 binding directly to the promoter region of the *P2rx4* gene and inducing *de novo* expression of P2X4R in microglia. Mice lacking *Irf5* do not show any upregulation of spinal P2X4R or any pain hypersensitivity after PNI. These

rsob.royalsocietypublishing.org *Open Biol.* **8**: 180154

rsob.royalsocietypublishing.org  Open Biol. 8: 180154

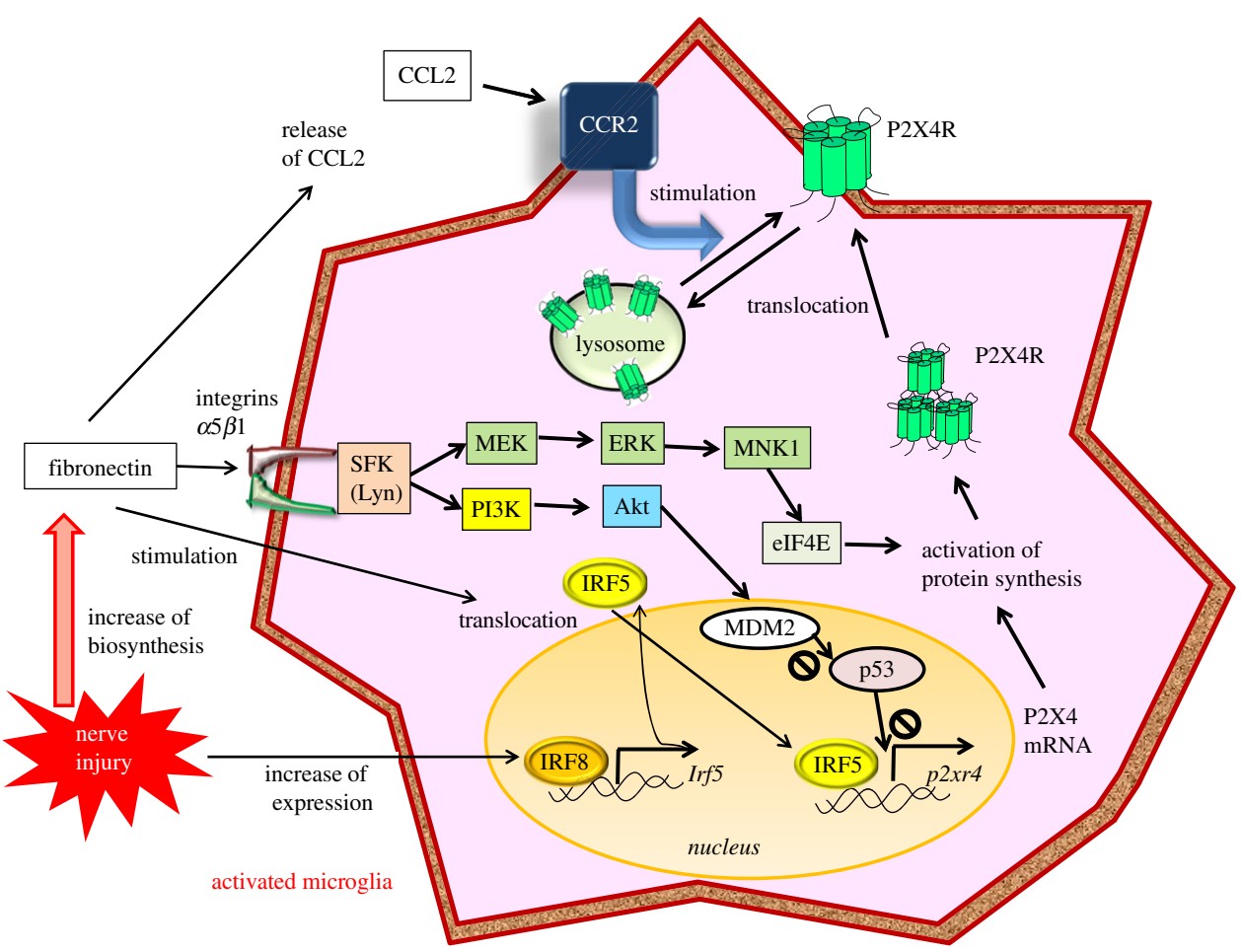

**Figure 3.** Increased P2X4R expression after nerve injury via the actions of fibronectin in activated microglia. Fibronectin increases after nerve injury and binds to $\alpha5\beta1$ integrin on microglial cells, resulting in the activation of phosphatidylinositol 3-kinase (PI3 K)-Akt and mitogen-activated protein kinase (MAPK) kinase-extracellular signal-regulated kinase (MEK-ERK) signalling cascades through the action of the tyrosine kinase Lyn. Signalling through the PI3 K-Akt pathway induces the degradation of p53 via MDM2 and enhances P2X4R gene expression. Activated MEK-ERK signalling enhances eukaryotic translation initiation factor 4E (eIF4E) phosphorylation via activated MAPK-interacting protein kinase-1 (MNK1), which may lead to the enhancement of P2X4R translation. Thus, fibronectin enhances the expression of functional P2X4Rs (centre of this figure). The subcellular localization of P2X4R is restricted to lysosomes and is translocated into the cell membrane by stimulation. Fibronectin increases the release of CCL2 from activated microglia, and CCL2 stimulates the CC-chemokine receptor CCR2, resulting in the stimulation of P2X4R trafficking to the cell surface from lysosomes of microglia (upper side of the figure). Interferon 8 (IRF8) expression is markedly enhanced in microglia after PNI. IRF8 binds directly to the promoter loci of IRF5 and activates IRF5 transcription, increasing IRF5. Fibronectin stimulates IRF5 translocation from the cytosol to the nucleus. IRF5 induces *de novo* expression of P2X4R by binding directly to the promoter region of the *p2rx4* gene.

findings suggest that a transcriptional axis from IRF8 to IRF5 contributes to the activation of spinal microglia with P2X4R overexpression and the expression of tactile allodynia in neuropathic pain after PNI.

Another study showed that fibronectin is elevated in the dorsal horn after PNI and evokes an increase in mRNA and P2X4R protein in primary cultured microglial cells [32]. In addition, it was reported that an inhibitor of fibronectin/integrin signalling reduces the expression of P2X4R and tactile allodynia after PNI and that intrathecal administration of fibronectin evokes tactile allodynia in naive animals but does not produce allodynia in P2X4R-deficient mice [33]. Moreover, it was reported that fibronectin fails to cause the upregulation of P2X4R gene expression in microglial cells lacking Lyn [34]. Lyn, an important molecule of fibronectin/integrin signalling in microglia, is reported to be the main Src-family kinase (SFKs) in spinal microglia among the five Src-family members (Src, Fyn, Lck, Yes and Lyn). The level of Lyn increases after PNI, and tactile allodynia, as well as the upregulation of P2X4R, are suppressed in spinal microglia in mice lacking Lyn after PNI [34].

Two cascades of intracellular signalling are activated by Lyn tyrosine kinase [35]. One is a pathway via phosphatidyl-inositol 3-kinase (PI3 K)−Akt, and the other is a pathway via mitogen-activated protein kinase (MAPK) kinase (MEK)-extracellular signal-regulated kinase (ERK) [36]. Activation of the PI3 K−Akt pathway causes the degradation of p53 in a proteasome-dependent manner, which in turn leads to an enhancement of P2X4 gene expression, because p53 is inhibiting the expression of the P2X4 gene. Activated MEK-ERK signalling by fibronectin stimulates phosphorylation of eukaryotic translation initiation factor 4E (eIF4E) through the activation of MAPK-interacting protein kinase-1. Thus, this pathway may play a role in regulating P2X4 expression in microglia. Inhibition of SFK reportedly suppresses ERK activity in spinal microglia [37]. These results indicate that Lyn might be a key kinase in the P2X4R upregulation in microglia after PNI (figure 3).

There is an extremely interesting finding that the subcellular localization of P2X4R is restricted to lysosomes around the perinuclear region, and these P2X4 receptors are translocated into the cell membrane by stimulation [38]. Stimulation by

fibronectin increases the release of CCL2 from activated microglia, and CCL2 stimulates the CCR2 CC-chemokine receptor on microglial cell membranes, resulting in the stimulated trafficking of P2X4R to the cell surface from lysosomes in the microglia [39] (figure 3). Thus, fibronectin is a key regulator of P2X4R expression through various pathways.

# 6. Conclusion

In recent decades, an accumulating body of literature has provided evidence for the crucial role of microglia in neuropathic pain and our understanding of the molecular and cellular basis of neuropathic pain. However, some questions require clarification, especially when faced with the reality that approximately 40 molecules are selectively upregulated in spinal microglia following PNI [21], resulting in the hypothesis that each may be independently capable of contributing to neuropathic pain. This review describes the recent advances in the understanding of mechanisms of neuropathic pain, with a focus on P2X4 receptor functions in spinal microglia after PNI. Spinal microglia also express other purinergic receptors, including P2X7, P2Y12 and P2Y6, which show interesting functions related to neuropathic pain. The role of purinergic microglial signalling in the mechanisms of neuropathic pain provides crucial insights in its pathogenesis and suggests potential strategies for developing new treatments for neuropathic pain.

Data accessibility. This article has no additional data.

Competing interests. I declare I have no competing interests.

Funding. This work was supported partly by JSPS KAKENHI grant no. 25117013 and the Core Research for Evolutional Science and Technology (AMED-CREST) programme from Japan Agency for Medical Research and Development.

Acknowledgements. I thank Prof. Makoto Tsuda, Dr Takahiro Masuda and many students for the experiments. I thank Trent Rogers, PhD, from Edanz Group (www.edanzediting.com/ac) for editing a draft of this manuscript.

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
