## [Reviewer comments · Open Biology]

Review History

RSOB-18-0154.R0 (Original submission)

Review form: Reviewer 1

Recommendation

Accept with minor revision (please list in comments)

Are each of the following suitable for general readers?

- a) **Title**
Yes
- b) **Summary**
Yes
- c) **Introduction**
Yes

Is the length of the paper justified?

Yes

Should the paper be seen by a specialist statistical reviewer?

No

Is it clear how to make all supporting data available?

Not Applicable

Is the supplementary material necessary; and if so is it adequate and clear?

Not Applicable

Do you have any ethical concerns with this paper?

No

Comments to the Author

This is an interesting review written by the main specialist of the field. I have only a very few remarks to make:

1. The English of the MS should be improved. Especially in the first half of it there are many grammatical inconsistencies.
2. P.3, l.4. It could be mentioned that the central C-fiber terminals of DRG neurons project to the substantia gelatinosa (Layer II). Then interneurons project to the Layer I of the spinal cord.
3. Fig. Legends are missing. References to the Figs. should be made in the text.

Review form: Reviewer 2

Recommendation

Major revision is needed (please make suggestions in comments)

Are each of the following suitable for general readers?

- a) **Title**
Yes
- b) **Summary**
Yes
- c) **Introduction**
Yes

Is the length of the paper justified?

Yes

Should the paper be seen by a specialist statistical reviewer?

No

Is it clear how to make all supporting data available?

Not Applicable

Is the supplementary material necessary; and if so is it adequate and clear?

Not Applicable

Do you have any ethical concerns with this paper?

No

Comments to the Author

This is a very interesting and timely review on the role of microglia in the development of neuropathic pain, written by one of the pioneers in this field who significantly contributed (and still contributes) to the shift from a "neuron-centric" view of neuropathic pain to a more integrated approach. The concept of "microgliopathic" pain is extremely interesting as well, since microglia can trigger pain itself, and could hopefully represent an innovative target for the development of effective drugs.

Specific comments:

1. The paper would benefit from a careful proofreading due to several grammar and syntax errors and to several unclear sentences. Indeed, in the two last paragraphs of section #6 the statement "It was reported that..." is repeated at least 5 times in few lines; this makes the quite repetitive. Just few examples of unclear and/or incorrect sentences:

- Summary:

o "...chronic pain is lacking the physiological significance but rather of confusion of nerve functions"

- Introduction:

o "Neuropathic pain...is heavy clinical problem..."

o "Patients more than 20 million in the worldwide..."

o "...body of evidences [uncountable] showing that spinal microglia activated in response..."

- Section #4:

o "This report made a big stream as glial pain into the pain research."

2. I would change the title of Section #5 since mechanisms of evoking mechanical allodynia have been discussed also in the previous section. It could be changed to "Role of extracellular nucleotides in mechanical allodynia" or something similar and more connected to the topic of the section.

3. The reader is a bit confused by the background leading to Section #6 and to the development of the concept of "Microgliopathic pain". Maybe I got it wrong, but it is initially stated that this is abnormal pain derived by activated microglia without nerve damage. Nevertheless, after the first paragraph, the author discusses the mechanisms of microglia activation following PNI, which is a type of nerve damage. Please, try to better clarify this point.

4. Figure legends are missing.

Decision letter (RSOB-18-0154.R0)

03-Oct-2018

Dear Professor Inoue

We are pleased to inform you that your manuscript RSOB-18-0154 entitled "A state-of-the-art

prospect of microgliopathic pain" has been accepted by the Editor for publication in Open Biology. The reviewer(s) have recommended publication, but also suggest some minor revisions to your manuscript. Therefore, we invite you to respond to the reviewer(s)' comments and revise your manuscript.

Please submit the revised version of your manuscript within 14 days. If you do not think you will be able to meet this date please let us know immediately and we can extend this deadline for you.

- 1) A text file of the manuscript (doc, txt, rtf or tex), including the references, tables (including captions) and figure captions. Please remove any tracked changes from the text before submission. PDF files are not an accepted format for the "Main Document".
- 2) A separate electronic file of each figure (tiff, EPS or print-quality PDF preferred). The format should be produced directly from original creation package, or original software format. Please note that PowerPoint files are not accepted.
- 3) Electronic supplementary material: this should be contained in a separate file from the main text and meet our ESM criteria (see <http://royalsocietypublishing.org/instructions-authors#question5>). All supplementary materials accompanying an accepted article will be treated as in their final form. They will be published alongside the paper on the journal website and posted on the online figshare repository. Files on figshare will be made available approximately one week before the accompanying article so that the supplementary material can be attributed a unique DOI.

Online supplementary material will also carry the title and description provided during submission, so please ensure these are accurate and informative. Note that the Royal Society will not edit or typeset supplementary material and it will be hosted as provided. Please ensure that the supplementary material includes the paper details (authors, title, journal name, article DOI). Your article DOI will be 10.1098/rsob.2016[last 4 digits of e.g. 10.1098/rsob.20160049].

- 4) A media summary: a short non-technical summary (up to 100 words) of the key findings/importance of your manuscript. Please try to write in simple English, avoid jargon, explain the importance of the topic, outline the main implications and describe why this topic is newsworthy.

Images

Data-Sharing

It is a condition of publication that data supporting your paper are made available. Data should be made available either in the electronic supplementary material or through an appropriate repository. Details of how to access data should be included in your paper. Please see <http://royalsocietypublishing.org/site/authors/policy.xhtml#question6> for more details.

Data accessibility section

Sincerely,

The Open Biology Team
<mailto:openbiology@royalsociety.org>

Reviewer(s)' Comments to Author:

Referee: 1

Comments to the Author(s)

This is an interesting review written by the main specialist of the field. I have only a very few remarks to make:

1. The English of the MS should be improved. Especially in the first half of it there are many grammatical inconsistencies.
2. P.3, l.4. It could be mentioned that the central C-fiber terminals of DRG neurons project to the substantia gelatinosa (Layer II). Then interneurons project to the Layer I of the spinal cord.
3. Fig. Legends are missing. References to the Figs. should be made in the text.

Referee: 2

Comments to the Author(s)

This is a very interesting and timely review on the role of microglia in the development of neuropathic pain, written by one of the pioneers in this field who significantly contributed (and still contributes) to the shift from a "neuron-centric" view of neuropathic pain to a more integrated approach. The concept of "microgliopathic" pain is extremely interesting as well, since microglia can trigger pain itself, and could hopefully represent an innovative target for the development of effective drugs.

Specific comments:

1. The paper would benefit from a careful proofreading due to several grammar and syntax errors and to several unclear sentences. Indeed, in the two last paragraphs of section #6 the statement "It was reported that..." is repeated at least 5 times in few lines; this makes the quite repetitive. Just few examples of unclear and/or incorrect sentences:

- Summary:

o "...chronic pain is lacking the physiological significance but rather of confusion of nerve functions"

- Introduction:

o "Neuropathic pain...is heavy clinical problem..."

o "Patients more than 20 million in the worldwide..."

o "...body of evidences [uncountable] showing that spinal microglia activated in response..."

- Section #4:

o "This report made a big stream as glial pain into the pain research."

2. I would change the title of Section #5 since mechanisms of evoking mechanical allodynia have been discussed also in the previous section. It could be changed to "Role of extracellular nucleotides in mechanical allodynia" or something similar and more connected to the topic of the section.

3. The reader is a bit confused by the background leading to Section #6 and to the development of the concept of "Microgliopathic pain". Maybe I got it wrong, but it is initially stated that this is abnormal pain derived by activated microglia without nerve damage. Nevertheless, after the first paragraph, the author discusses the mechanisms of microglia activation following PNI, which is a type of nerve damage. Please, try to better clarify this point.

4. Figure legends are missing.

Author's Response to Decision Letter for (RSOB-18-0154.R0)

See Appendix A.

Decision letter (RSOB-18-0154.R1)

30-Oct-2018

Dear Professor Inoue

We are pleased to inform you that your manuscript entitled "A state-of-the-art prospect of microgliopathic pain" has been accepted by the Editor for publication in Open Biology.

You can expect to receive a proof of your article from our Production office in due course, please

check your spam filter if you do not receive it within the next 10 working days. Please let us know if you are likely to be away from e-mail contact during this time.

Sincerely,

The Open Biology Team
mailto: openbiology@royalsociety.org

Appendix A

28 October, 2018

Dear The Open Biology Team

re: Manuscript RSOB-18-0154

Thank you for your decision letter of 03-Oct-2018. I am very pleased that my manuscript is acceptable in principle and I have now revised the manuscript according to referee comments.

As you will see in the attached revised manuscript and in my point-by-point responses to referee comments, I have now addressed all of these points.

I am now submitting a revised version of the review manuscript. I appreciate very much your help and guidance with this manuscript.

Sincerely yours,

Kazuhide Inoue, Ph.D.

Department of Molecular and System Pharmacology, Graduate School of
Pharmaceutical Sciences, Kyushu University

3-1-1 Maidashi, Higashi-ku, Fukuoka 812-8582, JAPAN

Phone & Fax: 81-92-642-4729, E-mail: inoue@phar.kyushu-u.ac.jp

The point-by-point responses to referee comments

To Referee: 1

Comments to the Author(s)

Q1. The English of the MS should be improved. Especially in the first half of it there are many grammatical inconsistencies.

A: I am very sorry for my poor English. I asked Trent Rogers, Ph.D., from Edanz Group (www.edanzediting.com/ac) for editing a draft of this manuscript. I think the English has been improved.

Q2. P.3, l.4. It could be mentioned that the central C-fiber terminals of DRG neurons project to the substantia gelatinosa (Layer II). Then interneurons project to the Layer I of the spinal cord.

A: Thank you very much for your suggestion. As you mentioned, C-fiber terminals of DRG neurons project to the substantia gelatinosa (Layer II). And also there are reports indicating that C-fiber terminals of DRG neurons project to Lamina I. Then, I use “superficial layer” in the text as below because “superficial layer” includes Lamina I and II. “These spikes conduct to the central ends of these DRG neurons, and transmit to secondary sensory neurons in the dorsal horn superficial layer” This change does not cause essential misunderstandings.

3. Fig. Legends are missing. References to the Figs. should be made in the text.

A: I am very sorry for my mistake that is very embarrassing. It was completely dropped out in several corrective processes. I have attached Fig.Legends in revised version.

To Referee: 2

Comments to the Author(s)

Specific comments:

Q1. The paper would benefit from a careful proofreading due to several grammar and syntax errors and to several unclear sentences. Indeed, in the two last paragraphs of section #6 the statement “It was reported that...” is repeated at least 5 times in few lines; this makes the quite repetitive. Just few examples of unclear and/or incorrect sentences:

A: I am very sorry for my poor English. I asked Trent Rogers, Ph.D., from Edanz Group (www.edanzediting.com/ac) for editing a draft of this manuscript. I think the English has been improved.

Q2. I would change the title of Section #5 since mechanisms of evoking mechanical allodynia have been discussed also in the previous section. It could be changed to “Role of extracellular nucleotides in mechanical allodynia” or something similar and more connected to the topic of the section.

A: Thank you very much for your suggestion. Based on your remarks, I have reviewed all titles including Section #5 and the title of the paper. I changed the title of the paper from "A state-of-the-art prospect of microgliopathic pain" to “A state-of-the-art perspective of microgliopathic pain” as more appropriate. In Section #5, I firstly mentioned the mechanism of evoking mechanical allodynia. In previous section, I only mentioned the activation of microglia that results in neuropathic pain but not the mechanism. Therefore, the title of Section #5 seems to be appropriate as it is.

Q3. The reader is a bit confused by the background leading to Section #6 and to the development of the concept of “Microgliopathic pain”. Maybe I got it wrong, but it is initially stated that this is abnormal pain derived by activated microglia without nerve damage. Nevertheless, after the

first paragraph, the author discusses the mechanisms of microglia activation following PNI, which is a type of nerve damage. Please, try to better clarify this point.

A: Thank you very much for your suggestion. Based on your remarks, I have changed several sentences in the text.

Before: “It has been reported that neuropathic pain is caused by several treatments associated with activated microglia even in normal animals [17,19,23-26].”

Revised text; Neuropathic pain is caused by several conditions after PNI as mentioned before. Recently several reports indicated that abnormal activated microglia are able to cause neuropathic pain even in in naïve animals [17,19,23-26].

Q4. Figure legends are missing.

A: I am very sorry for my mistake that is very embarrassing. It was completely dropped out in several corrective processes. I have attached Fig.Legends in revised version.